# A Review of Bioactive Compounds and Antioxidant Activity Properties of *Piper* Species

**DOI:** 10.3390/molecules27196774

**Published:** 2022-10-10

**Authors:** Nono Carsono, Sefren Geiner Tumilaar, Dikdik Kurnia, Diding Latipudin, Mieke Hermiawati Satari

**Affiliations:** 1Plant Breeding Laboratory, Faculty of Agriculture, Universitas Padjadjaran, Sumedang 45363, Indonesia; 2Department of Chemistry, Faculty of Mathematics and Natural Science, Universitas Padjadjaran, Sumedang 45363, Indonesia; 3Department of Animal Nutrition, Faculty of Animal Husbandry, Universitas Padjadjaran, Sumedang 45363, Indonesia; 4Department of Oral Biology, Faculty of Dentistry, Universitas Padjdajaran, Sumedang 45363, Indonesia

**Keywords:** antioxidant, *Piper amalago*, *Piper betle*, *Piper hispidum*, *Piper longum*, *Piper umbellatum*

## Abstract

Antioxidants are compounds that are able to inhibit the negative effects that come from free radicals. The phenomenon of imbalanced antioxidant production and the accumulation of free radicals in cells and tissues can cause oxidative stress. Excessive free radicals that enter the body cannot be warded off by endogenous antioxidant compounds so that the required antioxidant compounds can come from the outside, which helps in the performance of endogenous antioxidants. Antioxidants that come from outside consist of synthetic and natural antioxidants; however, synthetic antioxidants are not an option because they have toxic and carcinogenic effects. Therefore, the use of natural ingredients is an alternative method that is needed to create a new natural antioxidant compound. *Piper* species are being considered as possible medicinal plants for the development of new sources of antioxidants. Several studies have been carried out starting from the extract levels, fractions, and compounds of the *Piper* species, which showed good antioxidant activity. Currently, some of these plants are being used as ingredients in traditional medicines to treat allergies, toothaches, and coughs. This review examines the distribution, botanical data, pharmacology, especially antioxidant activity, and the compounds contained in five *Piper* species, namely *Piper amalago* L., *Piper betle* L., *Piper hispidum* Sw., *Piper longum* L., and *Piper umbellatum* L.

## 1. Introduction

Antioxidants are molecules that delay or inhibit the oxidation process of unstable molecules and can prevent cell damage caused by free radicals [1,2]. Free radicals are highly reactive, unstable, and short-lived molecules because they have unpaired electrons; therefore, they tend to bind with other molecules to attain stability [3]. Molecules that are attacked become free radicals and can cause cell damage [4]. The overproduction of free radicals and insufficient production of antioxidants may result in oxidative stress [5,6]. Oxidative stress can cause several diseases such as cancer, stroke [7], diabetes, and myocardial infarction [8]. Antioxidants from the body (endogenous) are needed to prevent this condition. However, endogenous antioxidants such as the enzymes Superoxide dismutase (SOD), Catalase (CAT), and Glutathione peroxidase (GSHPx) cannot scavenge the overproduction of free radicals [9]. Antioxidant compounds from outside the body (exogenous antioxidants) can assist in the function of these enzymes. Natural antioxidants can be a source of compounds that can scavenge free radicals [10]. Several studies have reported that the *Piperaceae* have good antioxidant activity [11,12,13]. 

*Piperaceae* plants contain more than 10 genera and more than 1500 species [14]. These piper species are scattered in various places, and about 700 species are distributed in tropical America, 300 species occur in the Asian tropics, and another 15 species are found in Africa [15]. These plants have been widely used in traditional medicine in China, Indonesia, India, and Korea [16] and have many biological activities such as antifungal [17], antioxidant [18], anti-inflammatory [19], antimicrobial [20], and antidiabetic [21].

Bioactive compounds contained in piper species can be used as a source of new drug targets [22]. The bioactive compounds contained in the seeds, leaves, and stem bark are phenolic, tannins, saponins, alkaloids, flavonoids, glycosides, and terpenoids [23,24]. Amide Alkaloids are a unique compound present in *Piper* species; it is based on most of the compounds containing piper ring (pyridine) or pyrrole ring (pyrrollidine) [25]. Furthermore, some *Piper* species have a typical compound such as open-chain alkamides, cyclohexanamid, aristolactams, and ceramides. The main reason *Piper* species are used as a therapeutic alternative based on natural products is the content of bioactive compounds and various biological activities that have been reported in some of these plants. Therefore, the high content of compounds in *Piper* species can be used as the basis for drug discovery. This review aims to obtain information botany, ethnopharmacology, comprehensive compound with structured studies, and the antioxidant activity properties of five *Piper* species (*Piper amalago* L., *Piper betle* L., *Piper hispidum* Sw., *Piper longum* L., and *Piper umbellatum* L.) (see Appendix A).

## 2. Distribution, Botanical, Traditional Use, and Pharmacological Properties of Five *Piper* Species

*Piper amalago* L. is a plant spread in Brazil and Mexico. In Brazil, this plant is known by the name of “jaborandi-manso” and, in Mexico, this plant is commonly known as “kw’alaal its” [26]. This plant is included in a type of shrub that can reach a height of 2–7 m, the stem and leaves of which contain volatile oils [27]. *P. amalago* can relieve chest pain, and the leaves have been used in traditional medicine in Central America to treat digestive problems and heal burns, abscesses, boils, and insect bites. Chewed leaves are applied to bleeding cuts [28,29,30,31].

*P. amalago* has been used in folk medicine to prevent miscarriage and relieve pain in pregnant women and postpartum [32]. Recently, the aqueous extract from *P. amalago* leaves can help heal wounds on the thumbs of patients with type 2 diabetes mellitus for 15 days [33]. In addition, pharmacological studies have shown that *P. amalago* has biological activities like schistosomicidal [34], antimicrobial [35], anxiolytic [36], and anti-inflammatory [28].

*Piper betle* L., green Betel, is cultivated in Indonesia, East Africa, the Philippines, Malaysia, and India [37,38]. About a hundred varieties of this Betel plant are found worldwide—forty grow in Indonesia and thirty grow in Bangladesh [39]. This plant thrives in areas with an altitude of up to 900 m^2^ and is found in areas with high rainfall and hot and humid climates [40,41].

This plant is an evergreen and perennial plant that has a strong stem at the nodes, heart-shaped leaves that alternate and are 15–20 cm in length, and white catkin, with many small roots [42,43,44]. The roots can climb with the tip of the shoot and can reach 3–10 m; when young, the stems are light green and marked by short lines; however, when mature, they are sturdy, slightly flat, and the internodes are 12 cm long and 1.2 cm in diameter [39].

In Indonesia, raw Betel leaf is chewed to strengthen teeth and whiten teeth [45,46] while, in Sri Lanka, Betel is used in the traditional treatment of peptic ulcers [47]. Regarding ethnomedicines, this plant has been applied in traditional skin and eye medicine [48]. This plant is also used in other traditional medicines such as for the swelling of the gums, rheumatism, abdominal pain, coughs, colds, bronchial asthma, and constipation. Another purpose of using this plant is for the manufacture of insecticides, fish poisoning, perfumes, and oils [49,50].

*P. betle* leaves have been tested for several activities in vitro, such as antimicrobial [51], antidiabetic [52], neuroprotective [53], and antioxidant [54,55]. Abrahim et al. [56] reported that the ethyl acetate extract of *P. betle* leaves can inhibit the proliferation of MCF-7 cells as a chemotherapy agent for breast cancer treatment. 

*Piper hispidum* Sw. is an herbaceous plant that spreads in several tropical and subtropical countries in Central and South America [57]. This plant is known by the names “cordoncillo” (Mexico), “bayuoyo” (Cuba), and “jaborandi” (Brazil) [58,59]. This plant is a shrub that has alternating leaves, green cylindrical stems, and roots with sclereids in the parenchyma [60].

*P. hispidum* is commonly used by Nicaraguans to treat pain and wounds, treat urinary tract infections in Brazil, and regulate menstruation in Peru [61]. The flower parts of this plant are used topically to treat muscle pain. Meanwhile, the leaves of the plant are used by the people of Guatemala to treat female reproductive disorders and treat symptoms of skin leishmaniasis [62]. A decoction of *P. hispidum* leaf tea is useful for the treatment of malaria, and an infusion of *P. hispidum* leaves is combined with *P. aduncum* to treat stomach aches and colds in Jamaica [63]. This plant is popular in Ecuador for treating snakebites, insect bites, thrush, diarrhea, and conjunctivitis [30].

Several in vitro activity tests have been carried out on this plant. *P. hispidum* leaf extract has high antibacterial activity against *Enterococcus faecalis*, *Proteus mirabilis*, and *Candida albicans* bacteria [64]. The activity of this extract against MCF-7 cells showed an agonist or inhibitory effect in the formation of new cancer cells [65]. In addition, the pharmacological activity of *P. hispidum* is antioxidant, insecticidal, antiplasmodial, α-amylase, and antifungal activity [66,67].

*Piper longum* L. is referred to as “long pepper” in Javanese, Indian, and Indonesian. The word pepper comes from Sanskrit which means “pippali” [68]. This plant grows in hot areas in India and in some areas in Indonesia as well as in some subtropical areas such as Sri Lanka, America, and the Middle East [69,70]. This plant belongs to the category of vines with perennial woody roots and has heart-shaped leaves with unisexual flowers. The stems can grow from 0.6 to 0.9 m. The fruits, which grow in fleshy spikes 2.4–3.2 cm long and 5 mm thick, are blunt, oblong, and blackish green [71,72,73].

*P. longum* fruit is used as a spice after the drying process. It is also used as a remedy for diseases of the digestive tract and respiratory tract. During the middle ages, European and Chinese people used this plant as a spice and for curing various digestive ailments [74]. The most important aspect about this plant is that it was used in the traditional medicine system of the Indian tribes, namely Siddha, Ayurveda, and Unani [75,76]. *P. longum* fruit was used in Ayurveda to treat stomach ailments, leprosy, fever, and parasitic infections, while the roots were used to treat tumors and stomach enlargement [77,78].

Piperine is the main compound in *P. longum* and is used to treat bronchitis, dysentery, and insomnia [79,80]. *P. longum* has properties such as anticancer, antioxidant, anti-inflammatory, immunomodulatory, antiplatelet, analgesic, radioprotective, and antifertility [71,72,81].

*Piper umbellatum* L. is an herbaceous plant commonly known as “cow-foot”. It is native to the tropical rainforests of Africa, Amerika, India, and Nigeria. This plant is spread in all geographical areas in Brazil [82]. This plant thrives in cool, humid places with little light. It also grows in forest areas which usually reach 1.0–2.5 m [83,84]. The leaves form an almost circular shape with a dark-green top color and a grayish color on the underside of the leaf. The petiole is 6.5–30 cm long. This plant has small flowers with a width of 0.5–0.8 mm [85].

*P. umbellatum* is usually used for black soup or food seasoning in Nigeria. According to ethnopharmacology research in Cameroon, the roots of this plant are used to treat infertility and the leaves have been proven to treat anemia, genital infections, menstruation, and kidney disorders [86,87]. In Malaysia, the leaves of this plant are used in fish soup dishes because it is believed to have high nutritional value [88]. Every part of this plant is used for medicine in Brazil, such as constipation, colic, edema, and diarrhea. There have been at least 94 traditional medicines derived from this plant [30,89,90].

Based on a scientific literature study, the ethanol extract of the leaves of *P. umbellatum* has antioxidant [91], antifungal [92], anti-inflammatory [93], and anticancer [94] activity. da Silva et al. [95] investigated the activity of a hydroethanolic extract from *P. umbellatum* leaves, which was very effective in inhibiting the activity of *Enterococcus faecali*s, *Shigella flexneri*, and *Salmonella typhimurium*, with an MIC value of 12.5 μg/mL

## 3. Chemical Composition of Five *Piper* Species

### 3.1. Piper amalago L.

One of the main chemical constituents in *piper* species is an essential oil whose composition depends on the species and its growing habitat. In the *P. amalago*, the dominant compounds contained in the leaf extract of this plant are monoterpenes and sesquiterpenes [96,97,98]. The most abundant chemical fractions in the leaves of this plant are monoterpene hydrocarbons (33–34%), sesquiterpene hydrocarbons (27–37%), and oxygenated sesquiterpenes (27–31%) [32]. Research conducted by Santos et al. [99] stated that the main chemical components in *P. amalago* are α-cadinol (**1**) (4.96%), β-cedrene (**2**) (5.15%), germacrene d-4-ol (**3**) (5.54%), β-muurolene (**4**) (7.85%), (E)-nerolidol (**5**) (8.08%), dan β-phellandrene (**6**) (13.64%). Meanwhile, in research conducted by da Silva Mota et al. [100], α-amorphene (**7**) (25.7 %), *p*-cymene (**8**) (9.4 %), and (E)-methyl geranate (**9**) (7.8 %) were reported as the major constituents. de Ferraz et al. [97] showed β-caryophyllene (**10**) (4.69%), δ-elemene (**11**) (6.82%), zingiberene (**12**) (11.18%), zingiberone (**13**) (11.18%), and limonene (**14**) (20.52%) to be the main compounds. Table 1 shows the chemical composition of *P. amalago* found in the leaves, stem, root, flower, and fruit.

### 3.2. Piper betle L.

Based on proximate analysis, *P. betle* leaves contain water (85–90%), protein (3–3.5%), minerals (2.3–3.3%), carbohydrate (0.5–6.10%), essential oil (0.08–0.2%), vitamin A (1.9–2.9 mg/100 g), chlorophyll (0.01–0.25%), fiber (0.4–1.0%), calcium (0.2–0.5%) [102]. A preliminary phytochemical analysis of *P. betle* showed the presence of terpenes, phenols, saponin, alkaloids, amino acids, tannins, flavonoids, and steroids [21,37]. Moreover, the phytochemical analysis performed by Kaveti et al. [103] revealed that the *P. betle* water extract contains alkaloids, reducing sugars, saponins, tannins, and glycosides. Nowadays, bioactive compounds derived from *P. betle* have been widely studied using several instruments. 4-*ρ*-coumaroylquinic acid (**125**) and 3-*ρ*-coumaroylquinic acid (**126**) were identified using the HPLC/Electrospray Ionization Mass Spectrometric (ESI-MS) [104]. Furthermore, compounds such as 4-chromanol (**127**) (27.81%), phenol 2 methoxy 4-(-2-propenyl) acetate (**128**) (61.5%), and eugenol (**129**) (20.37%), using aqueous and ethanol extracts, were identified by gas chromatography–mass spectrometry (GCMS) [105]. Table 2 shows the compounds that were isolated from *P. betle* using several different types of extraction.

*P. betle* contains 0.08–0.2% essential oil, which is classified as aldehydes, phenylpropanoids, sesquiterpenes, and monoterpenes (Table 3). Table 3 shows GCMS data on the number of essential oils to *P. betle* compared from two different places, namely Bogor, Indonesia, and Varanasi, India. Major components of *P. betle* essential oil are phenolics such as chavibetol (**137**) (53.1%), safrole (**138**) (48.69%), and 2-allyl-6-methoxyphenol (**139**) (25.96%). Another study found several prime compounds including linalool (**38**), α-humulene (**73**), methyl eugenol (**140**), 1, 8-cineole (**29**), *p*-cymene (**8**), 4-Allyl-2-methoxy-phenolacetate (**141**), β-caryophyllene (**10**), α-terpinene (**142**), and methyl chavicol (**143**) [48,110,111].

### 3.3. Piper hispidum Sw.

Several secondary metabolites contained in *P. hispidum* that have been reported include amides, flavonoids, and butenolides. Recently, two amides have been isolated using the supercritical carbon dioxide method, namely (3*Z*,5*Z*)-N-isobutyl-8-(3′,4′-methylenedioxyphenyl)-heptadienamide (**157**) and *N*-[3-(6′-methoxy-3′,4′-methylenedioxyphenyl)-2(*Z*)-propenoyl] pyrrolidine (**158**) [114]. In another study, Ruiz et al. [115] succeeded in isolating a compound from the leaves of *P. hispidum*, namely piperamine (**159**) and piperine (**160**). The chalcones group of flavonoids was isolated from *P. hispidum*. 2′-hydroxy-3,4,4′,6′-tetramethoxychalcone (**161**), 2′-hydroxy-4,4′,6′-trimethoxychalcone (**162**), and 2′,3-dihydroxy-4,4′,6′-trimethoxychalcone (**163**) were successfully isolated from the same part of *P. hispidum* [116]. Ruiz et al. [115] successfully isolated and characterized 2′-hydroxy-3′,4′,6′-trimethoxychalcone (**164**) from the leaves extract of *P. hispidum*. Meanwhile, a new compound belonging to the butenolide group that was isolated from *P. hispidum* leaves was 9,10-methylenedioxy-5,6-Z-fadyenolide (**165**) [117].

Most of the essential oils identified from the leaves of *P. hispidum* belong to the monoterpenes and sesquiterpenes groups. *P. hispidum* essential oils also presented phenylpropanoids and alcohols. Table 4 shows GCMS data comparing the essential oil content identified by several researchers.

### 3.4. Piper Longum L.

Alkaloids, terpenoids, sterols, and essential oils are chemical compounds isolated from *P. longum*. The plant parts that have been studied to date are fruits, roots, leaves, and seeds [118]. Piperine is the most abundant compound in the fruit of *P. longum*. In addition, several compounds were isolated, including asarinine (**191**), pellitorine (**192**), retrofractamide A (**193**), piperlongumine (**194**), brachystamide A (**195**), and longamide A (**196**) [76,119]. Table 5 shows that there are compounds that have been isolated from several parts of *P. longum.*

The essential oils isolated from *P. longum* are shown in Table 6, along with their percentages. Forty-nine compounds representing the total essential oils were identified in the leaf, fruit, stem, and root of *P. longum.* This consisted of monoterpenes, aromatic ester, and sesquiterpenes.

### 3.5. Piper Umbellatum L.

In general, the bioactive compounds contained in *P. umbellatum* are alkaloids, flavonoids, sterols, and terpenoids (which are generally found in essential oils). Piperumbellactam A–D (226–229) are each classified as an alkaloid compound, which is a pure compound that has recently been discovered in the branch of plants [121]. Other bioactive compounds in *P. umbellatum* are shown in Table 7. Most of the essential oils belong to the monoterpenes and sesquiterpenes groups. Table 8 shows that there are forty-eight essential oil compounds that have been isolated from the leaves of *P. umbellatum.*

## 4. Antioxidant Properties

The chain of chemical reactions can take place well in life because there is oxygen from the air for the oxidation process, which provides energy in the form of ATP [123]. Furthermore, oxygen can be produced from the oxidation-reduction process in a living organism. This process also takes place in the transfer of electrons from one atom to another atom [124]. However, when the flow of electrons is problematic, it will cause unpaired single electrons to produce free radicals. Free radicals are molecules, ions, or atoms that have unpaired electrons in their outer orbitals; therefore, they are very active and react with other molecules by oxidizing or reducing other atoms. Free radicals are also known as reactive oxygen species (ROS) and reactive nitrogen species (RNS). In the human body, the main source of ROS and RNS is the mitochondria, which are the by-products of aerobic respiration [125,126]. Hydroxyl, superoxide anion, alkoxyl, peroxyl, and nitric oxide are oxygen-centered free radicals. Superoxide radicals and nitric oxide are less reactive than hydroxyl and alkoxyl radicals, which have a half-life of 1 and 10^−9^ s, respectively, which are two molecules that are highly reactive and can attack nearby molecules quickly and cause severe damage [127,128]. Briefly, ROS are formed from O_2_ which activates NADPH oxidase to produce superoxide anion radicals and is further converted to O_2_ and H_2_O_2_ by superoxide dismutase (SOD). However, the presence of increased ROS or excessive pro-oxidants in the body can cause oxidative stress. This also happens due to a lack of antioxidant production in the body [129,130]. Oxidative stress can cause damage to the function of biological cells; therefore, several diseases can occur, such as acute kidney failure, diabetes, atherosclerosis, preeclampsia, and hypertension [131]. Therefore, the rate of increase in ROS needs to be controlled by several preventive mechanisms such as increasing the number of antioxidants and detoxifying enzymes.

Antioxidants are molecules that can neutralize free radicals and can inhibit the oxidation process of other molecules or can be used as reducing agents. The main function of this antioxidant is to end the chain reaction of ROS formation [132]. Antioxidant defense mechanisms can occur in the form of oxidants scavenging, converting toxic free radicals into less toxic initiation, enhancing endogenous antioxidant defense systems, and reducing or preventing free radical production. All of these mechanisms work by protecting the body from oxidative stress [129]. Antioxidants can be produced in the body (endogenous) and come from outside in the form of dietary supplements (exogenous). Endogenous antioxidants or enzymatic antioxidants are the primary defense, while exogenous antioxidants or non-enzymatic antioxidants act as a secondary defense against ROS [133].

Enzymatic antioxidants can protect the body against oxidative attack because of their ability to decompose ROS. Superoxide dismutase (SOD), catalase (CAT), and glutathione Peroxidases (GSHPx) are the main enzymes involved in the defense against ROS [134]. SOD is found in prokaryotic and eukaryotic cells as a major defense against ROS. Iron and manganese are the main prosthetic groups in prokaryotes while copper, zinc, and manganese are present in eukaryotes [135]. The main function of this enzyme is to convert the superoxide radical anion into hydrogen peroxide (H_2_O_2_). In mammalian tissue, SOD is divided into three types, namely superoxide dismutase 1 (SOD1), which is found in the cytosol in the form of copper, superoxide dismutase 2 (SOD2), which is found in the mitochondria in the form of manganese, and extracellular superoxide dismutase. Recently, SOD1 is a major antioxidant enzyme in dealing with oxidative stress [136,137]. CAT is a heme enzyme group that catalyzes the breakdown of H_2_O_2_ in the water and protects cells against oxidative stress. CAT can transfer oxygen bound with H_2_O_2_ to other molecules. The decomposition of H_2_O_2_ used the peroxidative activity method (H_2_O_2_ + AH_2_ A + 2H_2_O) and the catalytic activity mode (2H_2_O_2_ + O_2_ + 2H_2_O). The breakdown of H_2_O_2_ by enzymes depends on the concentration of H_2_O_2_ and follows the first-order reaction [133,138,139]. GSHPx containing selenium plays a role in the detoxification mechanism. This enzyme catalyzes the reduction of H_2_O_2_ to water or alcohol. The mechanism of action of this enzyme starts from reduced glutathione, which functions as an effective electron donor because the thiol dioxide group becomes a sulfide bond (H_2_O_2_ + 2GSH → GS-SG + 2H_2_O) [140,141].

Exogenous antioxidants are further divided into synthetic antioxidants and natural antioxidants. These antioxidant compounds react with ROS to inhibit or eliminate the activity of these ROS [133]. Synthetic antioxidants interact with ROS through various mechanisms such as the binding of metal ions, deactivating singlet oxygen, converting radical into non-radical species, and absorbing UV radiation. Butylhydroxyanisol (BHA), butylhydroxytoluene (BHT), tert-butylhydroquinone (TBHQ), octyl gallate (OG), and propyl gallate (PG) are examples of synthetic antioxidants [142,143]. BHA is more effective than BHT due to the presence of two butyl groups which exhibit greater steric hindrance to other molecules. BHA is very effective in controlling the oxidation of short-chain fats [144]. When compared with TBHQ, BHA and BHT were less effective in inhibiting ROS activity. This is because the two para-hydroxyl groups in TBHQ are responsible for their antioxidant activity [144]. PG is known as a safe antioxidant because it can protect oil and food and rancidity resulting from the formation of peroxide [145]. Apart from being an antioxidant, PG can stabilize food and cosmetic ingredients. OG is a gallic acid and 1-octanol ester and is used as a food preservative. However, of these antioxidants, BHA and BHT are the most widely used antioxidants in the food industry [146]. The use of synthetic antioxidants has been limited and regulated by the Food and Drug Administration (FDA) because some synthetic antioxidants such as BHA and BHT have toxic and carcinogenic effects [143,147]. Synthetic antioxidants can cause DNA damage and induce premature aging [148]. BHA and BHT, in high doses, can cause adverse reactions such as liver damage, carcinogenesis, urticaria, eye problems, asthma, dermatitis, and angioedema [149]. PG can be added as a food additive to prevent bad odors that occur because it has higher chemical activity and suppresses the initiation of the chain oxidation of unsaturated fatty acids. However, the resulting sampling effect can result in toxicities or mutagenicities [150]. According to a study by Le Coz and Schneider [151], seven people experienced excessive reactions to BHA and BHT, such as headaches, drowsiness, pain radiating to the back, and asthma. Even when BHA and BHT were identified as presenting cross-reactivity with aspirin, twenty-one people experienced intolerance which caused dermatitis. These are some of the reasons that make natural compounds the main choice as natural antioxidants.

Natural antioxidants can strengthen endogenous antioxidant defenses in neutralizing and inhibiting the performance of ROS [152]. The main sources of these natural antioxidants can be found in fruits, leaves, seeds, vegetables, bark, and stems [145]. Phenolic is a secondary metabolite compound produced in various parts of the plant which has strong antioxidant activity. The mechanism of phenolic antioxidant activity includes hydrogen donation, metal ion chelation, singlet oxygen quenching, and free radical scavenging [153]. In addition, flavonoids also play an important role in reducing or preventing the toxicity caused by free radicals. Flavonoids protect the body from oxidative stress [129]. There are many compounds in plants that can inhibit the oxidation process; however, only a few are suitable for consumption due to safety concerns. They should not affect the odor, color, and taste, be easy to apply, stable in storage, must have an LD_50_ lower than 1000 mg/kg body weight, and must pass mutagenic, teratogenic, and carcinogenic tests. Currently, the development of natural antioxidants continues to be carried out because they have the same benefits as synthetic antioxidants but have no side effects [154,155]. Estévez and Ventanas [156] reported that essential oils from sage and rosemary exhibit antioxidant activity similar to BHT, prompting researchers to develop new raw materials from nature without compromising the quality of the resulting product. Currently, there are several benefits in the development of natural antioxidants such as green tea catechins that can inactivate the free radicals of skin collagen which can slow down skin aging [157], garlic as a plant for cardiovascular disease and cancer [158], and the consumption of red wine is good for atherosclerosis [159].

The determination of antioxidant activity in one sample can be performed with several models of antioxidant activity test methods. Currently, a test procedure has been developed to evaluate antioxidant activity in vitro. Antioxidant activity test methods cannot be compared with each other because they have different target mechanisms; therefore, each has advantages and disadvantages [160]. The method of determining antioxidant activity is divided into two main groups, namely hydrogen atom transfer (HAT) and single electron transfer (SET) methods. The HAT-based method is used to measure the ability of antioxidants to reduce free radical activity by donating hydrogen, while the SET-based method is used to detect the ability of antioxidants to reduce other compounds including free radicals [161]. 

SET-based methods are widely used for the in vitro testing of antioxidants. Several SET-based methods are 2,2-Diphenyl-1-picrylhydrazyl radical scavenging (DPPH), ferric ion reducing antioxidant power (FRAP),2,2-Azinobis 3-ethylbenzthiazoline-6-sulfonic acid radical scavenging (ABTS), cupric ions (Cu^2+^) reducing antioxidant power (CUPRAC), nitric oxide (NO) scavenging, superoxide anion (SA) radical scavenging and hydrogen peroxide (H_2_O_2_) scavenging assays [143,162].

Table 9 shows the antioxidant activity of the five piper species of the extract and fraction levels; however, compounds that have been isolated from these five plants are still underreported in their antioxidant activity.

At the compound level, Tabopda et al. [121] succeeded in isolating Piperumbellactam A–C (226-228) from *P. umbellactam* and tested its antioxidant activity using the DPPH method. Piperumbellactam A, B, and C have very strong antioxidant activity with IC_50_ values of 17.4, 8.1, and 13.7 μM, respectively. Furthermore, Eugenol (**129**), one of the compounds isolated from *P. betle*, has good antioxidant activity, with an IC_50_ value of 114.34 ± 0.46 g/mL when using the nitric oxide radical method [169].

## 5. Conclusions

*Piper amalago* L., *Piper betle* L., *Piper hispidum* Sw., *Piper longum* L., and *Piper umbellatum* L. are *Piper* species that have tremendous potential as a new source of natural-based medicines. This can be seen from the traditional use of this plant, which is used for dental and oral care. In addition, the content of extracts, fractions, and secondary metabolites has several biological activities, especially, in several methods, antioxidants. Based on this review, it is hoped that it can become a reference and guide in the development of several activities for the discovery of new drugs based on natural ingredients from this *Piper* species.

## Figures and Tables

**Table 1 molecules-27-06774-t001:** The chemical compounds of *P. amalago*.

Compounds	RI		Relative Area (%)					
	RI_Calc_ [100,101]	RI_Lit_ [100,101]	Unripe Fruit [101]	Ripe Fruit [101]	Flower [100]	Root [100]	Stem [100]	Leaf [100]
Ethyl isovalerate (**15**)	858	858			2.9		0.4	0.1
Tricyclene (**16**)	925	926		0.1			0.1	
α-Thujene (**17**)	929	930			0.8			1.8
α-Pinene (**18**)	933	939	0.7	3.6				
Camphene (**19**)	951	953		0.2				
Verbenene (**20**)	967	968						0.1
Sabinene (**21**)	974	975	1.3	3.0	0.8			0.2
β-Pinene (**22**)	980	979	0.2	0.6		1.2	3.6	1.8
*endo*-Norborneol (**23**)	985	986			1.5			
β-Myrcene (**24**)	988	991	1.8	2.6				
Pyrazine 2,3,5-trimethyl (**25**)	999	1000			0.4	0.4	0.4	0.2
δ-Carene (**26**)	1001	1002			1.1			0.3
α-Phellandrene (**27**)	1008	1005	0.7					
*p*-Cymene (**8**)	1024	1025	0.7	2.2	9.3	3.3	0.3	9.4
*o*-Cymene (**28**)	1025	1026			1.1	3.2	2.1	0.2
Limonene (**14**)	1029	1029	1.0	1.5	10.5	3.0	1.1	0.9
β-Phellandrene (**6**)	1032	1031	8.2	7.3				
1.8-Cineole (**29**)	1034	1033	0.3	0.1				
Acetyl pyridine (**30**)	1034	1034			0.1			
(Z)-β-Ocimene (**31**)	1037	1037	0.1	<0.1	0.3	2.1		0.4
γ-Terpinene (**32**)	1058	1062		<0.1				
*cis*-Sabinene hydrate (**33**)	1072	1068		<0.1				
Benzyl formate (**34**)	1077	1076					0.2	
*p*-Mentha-2,4(8)-diene (**35**)	1087	1086		<0.1				
Terpinolene (**36**)	1090	1089				2.4	1.6	
*p*-Cymenene (**37**)	1092	1091						3.4
Linalool (**38**)	1101	1098	2.0	1.4				
*cis*-Pinene hydrate (**39**)	1127	1121	0.06					
Pyrazine 3-methyl-2-isobutyl (**40**)	1137	1137				0.3	0.2	
*cis*-β-Terpineol (**41**)	1145	1144	0.04					
Menthol (**42**)	1173	1172				1.1	0.2	
Borneol (**43**)	1175	1165	0.1					
4-Terpineol (**44**)	1183	1177	0.3	0.2				
α-Terpineol (**45**)	1188	1189	0.2	0.5		1.3	1.2	0.1
Cryptone (**46**)	1191	1185	0.1	0.9				
*trans*-Dihydro carveol acetate (**47**)	1308	1307				1.3	0.2	
*iso*-Verbenol acetate (**48**)	1310	1310				0.7	3.2	5.1
(E)-Methyl geranate (**9**)	1325	1325			1.1	1.3	2.3	7.8
γ-Elemene (**49**)	1335	1339	0.4	0.5				
Presilphiperfol-7-ene (**50**)	1337	1337				3.3	0.4	
α-Cubebene (**51**)	1347	1351	0.2	0.1				
(Z)-β-Damascenone (**52**)	1364	1364				0.6	0.6	0.2
Cyclosativene (**53**)	1371	1371	0.1	<0.1		1.2	0.2	
Longicyclene (**54**)	1373	1374			1.2	2.2	2.1	0.2
α-Copaene (**55**)	1376	1376	3.0	0.7				
Silphiperfol-6-ene (**56**)	1379	1378			13.5			2.4
β-Bourbonene (**57**)	1384	1384	0.2	0.2				
β-Cubebene (**58**)	1388	1390	3.3	2.5				
β-Elemene (**59**)	1390	1391		0.5				
Sativene (**60**)	1392	1392				1.1	0.2	0.1
Longifolene (**61**)	1407	1408			1.2	2.3	6.6	3.0
α-Gurjunene (**62**)	1411	1410		0.5		4.4	1.1	
β-Funebrene (**63**)	1416	1415				1.3	0.3	
β*-*Caryophyllene (**10**)	1420	1418	2.6	2.7				
β-Cedrene (**2**)	1421	1421				0.4	1.1	0.2
β-Duprezianene (**64**)	1423	1423				1.0	1.0	0.4
β-Copaene (**65**)	1433	1432				1.2	1.2	
β-Gurjunene (**66**)	1434	1434		0.2		1.2	1.2	0.2
Aromadendrene (**67**)	1441	1441	0.1	<0.1		1.1	1.1	0.2
(Z)-β-Farnesene (**68**)	1443	1443				2.4	1.4	
Cedrane (**69**)	1444	1444			2.5			0.8
*epi*-β-Santalene (**70**)	1448	1447						0.5
*cis*-Muurola-3, 5-diene (**71**)	1450	1450				1.4	1.1	0.1
α-Himachalene (**72**)	1451	1451				1.6	0.5	
α-Humulene (**73**)	1456	1454	1.0	0.8				
α-Patchoulene (**74**)	1457	1457						3.9
*allo-*Aromadendrene (**75**)	1459	1460			7.0			0.1
Seychellene (**76**)	1460	1460		<0.1				
*cis*-Muurola-4(14),5-diene (**77**)	1468	1467			0.5	2.2	1.2	0.5
β-Acoradiene (**78**)	1471	1471				1.2	1.1	
γ-Himachalene (**79**)	1475	1476		2.3				
*trans*-Cadina-1(6),4-diene (**80**)	1477	1477				1.3	0.3	
β-Chamigrene (**81**)	1478	1478				1.3	1.3	
γ-Muurolene (**82**)	1479	1480	2.1		0.1			0.1
α-Amorphene (**7**)	1485	1485			2.0	14.4	23.3	25.7
Germacrene D (**83**)	1485	1485	2.0	1.0	18.5			0.2
*trans*-Muurola-4(14),5-diene (**84**)	1494	1494			0.6	1.6	1.5	1.1
Bicyclogermacrene (**85**)	1496	1496	9.1	3.0				
γ-Amorphene (**86**)	1496	1496			0.3			
Valencene (**87**)	1497	1496	0.3		1.1			
α-Muurolene (**88**)	1501	1500	1.5	<0.1	0.5	1.2	1.2	
Epizonarene (**89**)	1502	1502						0.3
Germacrene A (**90**)	1507	1503	0.9	0.7				
(Z)-α-Bisabolene (**91**)	1507	1507						4.8
Cuparene (**92**)	1505	1505			3.6			
γ-Cadinene (**93**)	1514	1514	0.9	0.9		1.3	2.3	
Myristicin (**94**)	1519	1519						0.2
*cis-*Calamenene (**95**)	1521	1521		<0.1				
Eugenyl acetate (**96**)	1523	1523						0.2
δ-Cadinene (**97**)	1523	1523	6.6	2.3		1.4	2.4	
α-Calacorene (**98**)	1542	1542		<0.1				
α-Elemol (**99**)	1549	1550						0.5
Italicene epoxide (**100**)	1549	1549			3.0			
(E)-Nerolidol (**5**)	1561	1564	14.2	19.9				
*epi*-Longipinanol (**101**)	1564	1564			4.8			
(Z)-Isoelemicin (**102**)	1570	1570						1.2
Germacrene d-4-ol (**3**)	1578	1576	10.3	12.7				
Spathulenol (**103**)	1578	1578				3.4	1.0	
Caryophyllene oxide (**104**)	1583	1581		0.7				0.5
Thujopsan-2-β-ol (**105**)	1587	1587						0.2
Globulol (**106**)	1588	1583		0.8				
Carotol (**107**)	1595	1595				1.4	1.1	
Guaiol (**108**)	1600	1601				1.2	1.1	0.1
1,10-*epi*-Cubenol (**109**)	1619	1619				1.9	1.5	1.5
*epi*-Cedrol (**110**)	1619	1619						1.5
Eremoligenol (**111**)	1631	1631				2.6	3.0	
γ-Eudesmol (**112**)	1632	1632						2.3
α-Acorenol (**113**)	1633	1633	1.2	2.1		2.3	3.1	0.6
β-Acorenol (**114**)	1637	1637		<0.1		2.4	1.4	
Cubenol (**115**)	1643	1643						6.2
*epi*-α-Cadinol (**116**)	1644	1640	6.1	4.9				
α-Muurolol (**117**)	1645	1646	2.6	2.1	5.0	6.3	9.3	
*epi*-α-Muurolol (**118**)	1646	1641	1.5					
α-Cadinol (**1**)	1658	1653	11.1	8.2				
*neo*-Intermedeol (**119**)	1661	1660				1.8	0.6	
Bulnesol (**120**)	1672	1672						1.3
α-Bisabolol (**121**)	1682	1683		<0.1				
*neo*-5-Cedranol (**122**)	1685	1685					0.1	
Oplopanone (**123**)	1737	1733	0.1	3.9				
Khusinol acetate (**124**)	1829	1816		<0.1				

RI*cal* = Retention indices relative (C_8_–C_26_) and apolar DB-5 column*;* Ri*lit* = Retention indices literature.

**Table 2 molecules-27-06774-t002:** List of isolated bioactive compounds identified from leaves extract of *P. betle*.

*P. betle* Leaves Extract	Compounds	References
Chloroform extract was identified using nuclear magnetic resonance (NMR)	1-*n*-dodecanyloxy resorcinol (**130**)	[15]
Ethanol extract using ultrasound-assisted extraction	Isoeugenol (**131**), eugenol (**129**), hydroxychavicol (**132**)	[106]
Soxhlet extraction	4-Allyl-1,2-diacetoxybenzene (**133**), 4-chromanol (**127**), hydroxychavicol (**132**), eugenol (**129**)	[107]
Hexane, ethyl acetate, and ethanol extract	Eugenol (**129**), 4-Allyl-1,2-diacetoxybenzene (**133**)	[108]
Crude aqueous extract	Benzeneacetic acid (**134**), eugenol (**129**), hexadecanoic acid (**135**), octadecanoic acid (**136**), hydroxychavicol (**132**)	[109]

**Table 3 molecules-27-06774-t003:** List of isolated bioactive compounds identified from leaves extract of *P. betle*.

Classification [112]	Compounds	Rt. (min.) [113]	Percentage (%) [113]	Rt. (min.) [48]	Percentage (%) [48]
Monoterpenes	α-Pinene (**18**)	3.874	0.34	9.6	0.09
	Camphene (**19**)			10.150	0.09
	Limonene (**14**)			13.100	0.28
	Sabinene (**21**)	4.290	0.83		
	γ-Muurolene (**82**)	3.761	0.34		
	β-Myrcene (**24**)	4.425	0.61		
	α-Phellandrene (**27**)	5.176	1.19		
	(*E*)-β-Ocimene (**144**)	5.403	0.10		
	γ-Terpinene (**32**)	5.722	0.99		
	α-Terpineol (**45**)	5.846	0.15		
	Terpinolene (**36**)	6.467	0.23		
	Linalool (**38**)	6.548	0.71		
Sesquiterpenes	Germacrene D (**83**)	21.020	0.75	34.251	2.85
	Germacrene B (**145**)			34.876	0.81
	γ-Muurolene (**82**)	20.912	2.84	33.926	1.27
	α-Humulene (**73**)	20.269	3.03	33.01	0.68
	β-Caryophyllene (**10**)	19.291	4.13	31.501	4.22
	β-Elemene (**59**)	18.378	0.61	30.176	0.24
	Ledane (**146**)			39.001	0.18
	Globulol (**106**)			40.126	0.12
	γ-Cadinene (**93**)	14.978	5.87	40.926	3.85
	α-Copaene (**55**)	17.957	0.83		
	Aromadendrene (**67**)	20.442	0.07		
	β-Selinene (**147**)	21.225	5.52		
	δ-Cadinene (**97**)	22.171	0.72		
	Caryophyllene Epoxide (**148**)	23.521	0.16		
Phenylpropanoids	Chavicol (**149**)	12.517	6.64	23.275	0.55
	Eugenol (**129**)	16.461	0.17	28.851	63.39
	Methyl eugenol (**140**)			30.426	0.21
	Acetyl eugenol (**150**)	21.522	9.62	35.826	14.05
	Isoestragole (**151**)	9.968	0.52		
	Isoeugenol (**131**)	17.114	20.71		
	Chavicol acetate (**152**)	16.120	17.75		
	Acetyl Isoeugenol (**153**)	21.603	3.96		
Aldehydes	Decanal (**154**)			20.975	0.18
	Undecanal (**155**)			30.576	0.43
	Phenyl acetaldehyde (**156**)			13.650	0.13

Rt = Retention Time

**Table 4 molecules-27-06774-t004:** Chemical composition of the essential oils of *P. hispidum*.

Compounds	Content Essential Oil of *P. hispidum* (%)		
	[57]	[58]	[64]
(*E*)-3-Hexen-1-ol (**166**)	1.0		
α-Thujene (**17**)		0.1	
α-Pinene (**18**)	6.6	1.2	15.3
Sabinene (**21**)	0.3		
β-Pinene (**22**)	12.0	1.1	14.8
Sylvestrene (**167**)	1.7		
(*Z*)-β-Ocimene (**31**)	1.4		
*trans*-Sabinene hydrate (**168**)	0.5		
Terpinen-4-ol (**169**)	0.3	1.0	
*iso*-Dihydrocarveol (**170**)	0.4		
Camphene (**19**)		0.1	0.4
6-Methyl-5-heptene-2-one (**171**)		0.5	
β-Myrcene (**24**)		1.2	0.9
α-Phellandrene (**27**)		0.5	
δ-Carene (**26**)			6.9
α-Terpinene (**142**)		14.0	
*p-*Cymene (**8**)		12.0	2.3
β-Phellandrene (**6**)		1.4	0.3
γ-Terpinene (**32**)		30.9	
Terpinolene (**36**)		7.3	
Verbenene (**20**)			0.5
Limonene (**14**)			2.3
1,3,8-*p*-Menthatriene (**172**)			0.2
(*E*)-Pinocarveol (**173**)			0.5
Aromadendrene (**67**)		1.4	
δ-Elemene (**11**)	0.3		
α-Cubebene (**51**)	0.4		
α-Humulene (**73**)		0.4	0.6
β-Elemene (**59**)			8.1
α-Copaene (**55**)	0.9	0.5	1.8
β-Cubebene (**58**)	0.5		
α-Gurjunene (**62**)	1.0		
β-Gurjunene (**66**)			0.8
γ-Gurjunene (**174**)			0.4
β-Caryophyllene (**10**)		5.3	6.2
Khusimene (**175**)	12.1		
α-*neo*-Clovene (**176**)	1.1		
β-Chamigrene (**81**)		1.6	
(*E*)-Ocimenone (**177**)			0.6
*cis*-Calamenene (**95**)			0.6
*cis*-β-Guaiene (**178**)	1.3		
β-Selinene (**147**)	1.0	8.1	
β-Bourbonene (**57**)			0.5
Valencene (**87**)	2.0		0.9
Myrtenol (**179**)			0.6
Viridiflorene (**180**)	3.4		1.0
α-Selinene (**181**)	3.6	9.0	
Epizonarene (**89**)		0.1	
γ-Cadinene (**93**)	13.2	0.4	0.8
δ-Cadinene (**97**)	6.3		
α-Muurolene (**88**)		0.2	
Selina-3,7(11)-diene (**182**)	0.6		
Germacrene A (**90**)			0.9
Germacrene B (**145**)	0.3		5.2
Ledol (**183**)	8.8		
Globulol (**106**)	0.7	1.2	
7-*epi*-α-selinene (**184**)		0.2	
Viridifloral (**185**)	3.0		
10-*epi*-Eudesmol (**186**)	1.1		
β-Eudesmol (**187**)			2.6
Hinesol (**188**)	0.3		
Cubenol (**115**)	4.2		
Selin-11-en-4-α-ol (**189**)	1.9	2.0	
*epi-*α-Cadinol (**116**)		0.5	
Guaiol acetate (**190**)	0.6		
Spathulenol (**103**)			5.0
Caryophyllene oxide (**104**)			7.8

**Table 5 molecules-27-06774-t005:** Other bioactive compounds of *P. longum* [68,118].

Compounds	Part Used	Compounds	Part Used
Piperrolein B (**197**)	Fruit	Cepharadione B (**203**)	Root
Dehydropipernonaline (**198**)	Fruit	Cepharadione A (**204**)	Root
Rosin (**199**)	Fruit	Methylpiperate (**205**)	Fruit
Piperchabaoside A (**200**)	Fruit	(+)-Diaeudesmin (**206**)	Seed
Sylvatine (**201**)	Seed	Aristolactam (**207**)	Root
Sesamin (**202**)	Seed	Piperoctadecalidine (**208**)	Fruit

**Table 6 molecules-27-06774-t006:** Other essential oils’ compositions of *P. longum* [120].

Compound	Content Essential Oil of *P. longum* (%)			
	Root	Stem	Fruit	Leaf
α-Pinene (**18**)	11.8	14.0	15.3	0.3
Camphene (**19**)	13.9	6.6	0.7	
β-Pinene (**22**)	26.4	34.8	43.1	1.6
β-Myrcene (**24**)	1.5	1.6	1.4	
α-Phellandrene (**27**)	0.2	0.4		
β-Phellandrene (**6**)	1.0	0.7	1.4	
Limonene (**14**)	6.3	10.3	9.6	0.7
1,8-Cineole (**29**)	0.9	0.8		
(*E*)-β-Ocimene (**144**)	0.7	0.6		
(*Z*)-β-Ocimene (**31**)	0.9		0.8	
Terpinolene (**36**)		0.4	0.5	
Linalool (**38**)		0.8	1.1	1.2
1-Methylhexyl acetate (**209**)			2.5	0.3
2-Nonanone (**210**)			1.3	
α-Terpineol (**45**)		0.4	0.5	
Decanal (**154**)				0.3
Bornyl acetate (**211**)	10.0	5.0		0.6
2-Undecanone (**212**)	0.8	1.0	2.9	2.0
Tridecane (**213**)	0.6	0.7		
δ-Elemene (**11**)	5.8	1.6		
γ-Elemene (**49**)				1.4
Terpinyl acetate (**214**)	1.1	0.5		
α-Copaene (**55**)				0.9
β-Cubebene (**58**)				2.3
β-Elemene (**59**)	0.1			1.4
Dodecanal (**215**)				0.4
β-Caryophyllene (**10**)	5.6	9.3	5.7	16.8
*cis*-β-Guaiene (**178**)				0.6
α-Gurjunene (**62**)		0.8		2.6
α-Humulene (**73**)	0.4	2.3		5.8
(*Z*)-α-Farnesene (**216**)				0.3
γ-Muurolene (**82**)				0.6
1-Pentadecene (**217**)	0.6			
2-Tridecanone (**218**)			0.9	0.4
Pentadecane (**219**)	5.0	0.7		
α-Muurolene (**88**)				1.8
β-Patchoulene (**220**)	0.7	0.8		
γ-Bisabolene (**221**)			0.6	
γ-Cadinene (**93**)				0.8
(*E*)-Nerolidol (**5**)	1.0	2.2	8.8	22.5
α-Elemol (**99**)				0.6
Germacrene A (**90**)				0.5
Caryophyllene oxide (**104**)	1.1	0.9		2.1
δ-Cadinol (**222**)	0.1			1.9
α-Cadinol (**1**)				0.5
β-Eudesmol (**187**)				3.3
α-Eudesmol (**223**)				0.7
Heptadecane (**224**)	0.4			-
9-Eicosyne (**225**)				1.5

**Table 7 molecules-27-06774-t007:** Chemical constituents of *P. umbellatum*.

Compounds	Plant Part	References
*N*-Hydroxyaristolam II (**230**)	Branches	[121]
Acacetin-7-O-β-D-galactopyranoside (**231**)	Branches	[121]
Rhoifolin (**232**)	Branches	[121]
Campestrol (**233**)	Aerial parts	[122]
β-Sitosterol (**234**)	Aerial parts	[122]
Stigmasterol (**235**)	Aerial parts	[122]
4-Nerolidylcatechol (**236**)	Roots	[121]
N-*p*-Coumaroyl tyramine (**237**)	Branches	[121]
N-*trans*-Feruloyl tyramine (**238**)	Branches	[121]

**Table 8 molecules-27-06774-t008:** List of essential oil of *P. umbellatum* [84].

Components	Ria	RIp	Percentage (%)
α-Pinene (**18**)	931	1020	0.2
6-Methyl-hept-5-en-2-one (**239**)	961	1344	0.1
Sabinene (**21**)	966	1127	0.1
β-Pinene (**22**)	972	1117	0.5
β-Myrcene (**24**)	981	1165	0.5
α-Phellandrene (**27**)	998	1171	0.2
δ-Carene (**26**)	1006	1154	0.1
*p*-Cymene (**8**)	1012	1277	0.3
Limonene (**14**)	1022	1207	12.5
1, 8-Cineole (**29**)	1022	1216	0.1
(*Z*)-β-Ocimene (**31**)	1025	1237	0.5
(*E*)-β-Ocimene (**144**)	1036	1254	0.2
γ-Terpinene (**32**)	1049	1250	0.1
Octan-1-ol (**240**)	1052	1552	0.1
Terpinolene (**36**)	1079	1289	0.1
Linalool (**38**)	1085	1552	41.1
Camphor (**241**)	1123	1527	0.1
4-Terpineol (**44**)	1163	1597	0.3
α-Terpineol (**45**)	1173	1703	1.1
Nerol (**242**)	1207	1798	0.1
Neral (**243**)	1210	1676	0.2
Geraniol (**244**)	1234	1842	0.1
Safrole (**138**)	1263	1861	0.1
Thymol (**245**)	1267	2193	4.3
α-Copaene (**55**)	1377	1497	0.1
β-Elemene (**59**)	1389	1595	0.5
β-Caryophyllene (**10**)	1421	1605	19.3
γ-Elemene (**49**)	1429	1635	0.2
*trans*-α-Bergamotene (**246**)	1435	1579	0.2
(*E*)-β-Farnesene (**247**)	1447	1669	0.2
α-Humulene (**73**)	1453	1676	1.3
γ-Muurolene (**82**)	1472	1693	0.2
Germacrene D (**83**)	1478	1703	1.2
β-Selinene (**147**)	1484	1715	0.9
Bicyclogermacrene (**85**)	1493	1728	0.9
(*E*)-α-Farnesene (**248**)	1496	1752	1.2
β-Bisabolene (**249**)	1501	1723	0.8
β-Sesquiphellandrene (**250**)	1508	1762	0.2
δ-Cadinene (**97**)	1516	1752	0.3
β-Elemol (**251**)	1535	2071	0.3
(*E*)-Nerolidol (**5**)	1548	2045	4.3
Caryophyllene oxide (**104**)	1573	1993	0.8
Humulene oxide II (**252**)	1597	2050	0.1
γ-Eudesmol (**112**)	1620	2176	0.1
β-Eudesmol (**187**)	1632	2238	0.1
α-Eudesmol (**223**)	1637	2228	0.1
(*E*)-Phytol (**253**)	2099	2613	1.8

RIa: retention indices measured on apolar (BP-1); Rip: retention indices measured on apolar (BP-20)

**Table 9 molecules-27-06774-t009:** Antioxidant activity test using several methods on five *Piper* species.

Five *Piper* Species	Sample	Test Methods Antioxidant IC_50_ (μg/mL)				References
		DPPH	ABTS	SA	NO	
*P. amalago* leaves	Ethanol	28.09				[66,163]
	Methanol	675	370			
	Dichloromethane	327	392			
*P. amalago* roots	Methanol	368	351			[66]
	Dichloromethane	371	509			
*P. betle* leaves	Methanol	16.33				[13,40,55,56,160,164]
		345.7		288.3	143.3	
	Ethyl acetate	40		48.3	52.3	
		23.25	79			
	Hexane	144.3			94.3	
	Aqueos			79.3	57.7	
		179.5				
	Ethanol	151.36				
		9.36	6.61			
*P. hispidum* leaves	Methanol	404	498			[66]
	Dichloromethane	391	158			
*P. hispidum* roots	Methanol	317	131			[66]
	Dichloromethane	263	164			
*P. longum* seeds	Ethanol	50			80	[165]
	Chloroform	6			76	
	Hexane	70			80	
	Ethyl acetate	54			80	
	Aqueos	19.5				
	Hydroethanol	26				
*P. longum* leaves	Methanol	149.92				[166]
*P. longum* fruits	Methanol	220.3		52.0		[167,168]
	Ethanol	89.8	238.4	482.3		
	Water	118.29	364.2	381.5		
*P. umbellatum* leaves	Methanol	312	423			[66]
	Dichloromethane	226	122			
*P. umbellatum* roots	Methanol	199	228			[66]
	Dichloromethane	19	102			

## Data Availability

The study did not report any data.

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
