# Peer review of "A Review of Bioactive Compounds and Antioxidant Activity Properties of Piper Species"

_molecules, 2022, doi:10.3390/molecules27196774_

Round 1
Reviewer 1 Report
The authors should include specific recent references and stress the importance of current therapeutic trends comparing them to contemporary methods. The pros and cons of synthetic adjunctive drugs are to be described briefly.
Authors in this review have to mention the reason for choosing and not choosing the therapeutic alternatives from different sources.
Author Response
- The authors should include specific recent references and stress the importance of current therapeutic trends comparing them to contemporary methods. The pros and cons of synthetic adjunctive drugs are to be described briefly.
Answer:
Thank you for taking the time to assess our manuscript. The authors have added recent references to the current trend of natural-based antioxidant therapy. We added the discussion on lines 354-366. A discussion of synthetic antioxidants has also been added to lines 335-345.
- Authors in this review have to mention the reason for choosing and not choosing the therapeutic alternatives from different sources.
Answer:
Thank you very much for the advice given. The authors have explained the importance of selecting Piper species as an alternative of antioxidant therapy in the introduction to this manuscript. In brief, the Piper species was chosen because it has a various bioactive compound and is reported to have certain biological activities.
Note: We have marked the revised part of the script with a yellow highlighter
Reviewer 2 Report
The manuscript "A Review of Bioactive Compounds and Antioxidant Activity Properties of Piper Species" by Carsono et al. leaves mixed impressions. On my opinion, this is due to the quite different quality of the two parts of the manuscript.
The text in chapters "2. Distribution, Botanical, Traditional Use and Pharmacological Properties of Five Piper Species" and "3. Chemical Composition of Five Piper Species" provides valuable compendium of phytochemical information about Piper spp. plants. However, these parts are somewhat piled with illustrative material - big tables and figures - which could partially be relocated to a supplement/supplements. The material would be even more useful if besides the chemical structures' drawings, the authors provide the structures in some machine-readable format (SMILES or SDF), in a supplement.
Some style amendments and typo corrections could further increase the quality of this sections:
l.68 "P. amalago has ability assuage chest pain,..." - to assuage ?
l.117 "Several in vitro activity tests have been carried out in this plant." - in this ?
l.176 "Meanwhile the study developed by da Silva Mota et al. [104]..." - developed ?
l.195,231,257,276,297 figure captions: "The structure chemical compound that obtained from ..." - of chemical compounds obtained from... ?
l.200 "...phytochemical analysis of P. betle showed the presence of terpenes, phenols, saponin, phenol, alkaloids, amino acids, tannins, flavonoid, and steroid [21, 38]." - mixed singular and plural forms ?
l.266 "The essential oils P. longum were shown in Table 6,..." - isolated from P. longum ?
l.284 "Most of essential oils belongs to monoterpenes..." - oils belong to ?
l.289, 291 table 8 caption "List of essential oil of P. umbellatum [88]" and legend: "* RIa: retention indices measured on apolar (BP-1); Rip: retention indices measured on polar (BP-20)" - list - maybe contents or compounds in? measured on apolar/polar - what?
Other part of the manuscript, in the chapter "4. Antioxidant Properties" describes only superficially the properties of the crude constituents (leaves, roots, fruits extracts) of Piper spp., though a simple google search of keywords "limonene antioxidant" provides a long list of papers concerning the subject (e.g. doi's 10.31024/ajpp.2018.4.6.25 , 10.1111/j.1742-7843.2009.00467.x ,
10.1016/j.fct.2015.04.015,
10.3390/antiox10060937 ). The bigger part of the text consists of general statements about oxidative stress and antioxidants' mechanism of action, not always expressed clearly:
l.311 Superoxide radicals and nitric oxide are less reactive than hydroxyl and alkoxyl radicals which have a half-life of 10 -9 s and 1 s, respectively, which are two molecules that are highly reactive and can attack nearby molecules quickly and cause severe damage [131-132]. timings are reversed in the "respectively" clause ?
l.326 Basically, antioxidant defense mechanisms can be ...converting toxic free radicals into less toxic initiation,... and increasing damaged molecules. ?
l.344 CAT is a heme enzyme group that catalyzes the breakdown of H 2 O 2 into the water and protects cells against oxidative stress. ... These four CAT polypeptide chains have more than 500 amino acids that can react with H 2 O 2 . ?
l.359 ...deactivating singlet oxygen scavenging ROS,... deactivating singlet oxygen, scavenging ROS ?
l.360 ...converting hydroperoxides into non-radical species,... hydroperoxides ARE non-radical.
l.363 ...presence of two tert-butyl groups which exhibit greater steric resistance to other molecules. - steric hindrance ?
l.374 ...makes natural compound compounds the main choice... duplicated "compounds"
l.380 ...hydrogendonation, metal ion cheating, ... hydrogen donation, metal ion chelation ?
l.382 ...flavonoids also play an important role in maintaining the toxicity caused by free radicals. ?
l.386 Determination of antioxidant activity cannot be determined based on one model of an antioxidant activity test method. ?
To that matter, in the Introduction and Abstract much more attention is payed to the not well expressed general statements about oxidative stress than to the succinct chapters 3 and 4 of the manuscript:
l.16 Excessive free radicals that enter the body cannot be warded off by endogenous antioxidant compounds so that the required antioxidant compounds can come from the outside which helps the performance of endogenous antioxidants.
l.18 Antioxidants that come from outside consist of synthetic and natural antioxidants, but synthetic antioxidants are not an option because they have toxic and carcinogenic effects.
l.20 Therefore, the use of natural ingredients is an alternative method that is needed to become a new natural antioxidant compound.
l.34 ...so they tend to bond with other molecules... - to bind ?
l.34 Molecules that are attacked becomes a free radical... - Molecule ? or become ?
l.42 ...compounds that can scavenging free radicals. - can scavenge ?
l.49 ...such as antifungal [17], antioxidant [18], antiinflamasi [19], antimicrobial [20], and antidiabetic [21]. - antiinflamasi ?
In this form, the manuscript is not well suited for publishing in "Molecules". Maybe authors would decide on splitting of the text, e.g. into "Bioactive compounds of Piper spp." and "Antioxidant properties of constituents of Piper spp.", but a major refinement of the antioxidant part of the text is necessary.
Author Response
1. The text in chapters "2. Distribution, Botanical, Traditional Use and Pharmacological Properties of Five Piper Species" and "3. Chemical Composition of Five Piper Species" provides valuable compendium of phytochemical information about Piper plants. However, these parts are somewhat piled with illustrative material - big tables and figures - which could partially be relocated to a supplement/supplements. The material would be even more useful if besides the chemical structures' drawings, the authors provide the structures in some machine-readable format (SMILES or SDF), in a supplement.
Answer:
Thank you for taking the time to assess our manuscript. We have modified the manuscript by transferring all images to supplementary materials. However, for all tables, we still wrote them in the manuscript because there are many important information that must be included in the main manuscript.
- Some style amendments and typo corrections could further increase the quality of this sections.
Answer:
Thank you for the advice given. We apologize for any writing errors. We have changed the writing error according to the suggestions given by the reviewer. The results of the changes have been marked in the manuscript.
- Other part of the manuscript, in the chapter "4. Antioxidant Properties" describes only superficially the properties of the crude constituents (leaves, roots, fruits extracts) of Piper, though a simple google search of keywords "limonene antioxidant" provides a long list of papers concerning the subject.
Answer:
Thank you for your comment. The list of compounds that have been recorded in the table is indeed derived from five Piper species. All of these compounds have discussed a lot about antioxidant activity of each compound, but not many studies have reported the antioxidant activity of these compounds which have been isolated especially from Piper species. Therefore, we only list and discuss a few compounds that have been isolated from Piper species that have antioxidant activity. We have added the discussion on lines 386-392.
- The bigger part of the text consists of general statements about oxidative stress and antioxidants' mechanism of action, not always expressed clearly
Answer:
Thank you for pointing this out. We apologize for the unclear writing. The authors have changed and clarified this error in the manuscript as suggested.
- To that matter, in the Introduction and Abstract much more attention is payed to the not well-expressed general statements about oxidative stress than to the succinct chapters 3 and 4 of the manuscript.
Answer:
Thank you for the advice given. The authors have added discussion and emphasis on oxidative stress to the abstract and introduction.
- In this form, the manuscript is not well suited for publishing in "Molecules". Maybe authors would decide on splitting of the text, e.g. into "Bioactive compounds of Piper" and "Antioxidant properties of constituents of Piper spp.", but a major refinement of the antioxidant part of the text is necessary
Answer:
Thank you for the advice given. Currently, the authors are still combining the discussion on “Bioactive compounds of Piper species” and “Antioxidant properties of constituents of Piper species” because in this review we emphasize the bioactive compounds and antioxidant activity of Piper species. The authors have added a discussion of the Antioxidant properties section.
Note: We have marked the revised part of the script with a yellow highlighter
Reviewer 3 Report
Authors have done a Review on Bioactive Compounds and Antioxidant Activity Properties of Piper Species. It is a well-written review they have discuss all the structures of different species of Piper and further elaborated its antioxidant properties. This manuscript can be accepted in its present form.
Author Response
1. Authors have done a Review on Bioactive Compounds and Antioxidant Activity Properties of Piper It is a well-written review they have discuss all the structures of different species of Piper and further elaborated its antioxidant properties. This manuscript can be accepted in its present form.
Answer:
Thank you for taking the time to assess our manuscript and gave your opinion to our manuscript. We are grateful that the reviewers have accepted our manuscript. Hopefully this review can be useful in the development of further research.
Round 2
Reviewer 2 Report
Relocation of figures to the supplementary material greatly improved the readability of the manuscript.Tables are OK in the main text.
Additional spell-checking and style corrections are necessary (thе problems marked in the review were in no way an exhaustive list).
Finally, I still would recommend a splitting of the text into "Bioactive compounds..." and "Antioxidant properties..." since not much was done to improve the antioxidant part in ch. 4.
Author Response
- Relocation of figures to the supplementary material greatly improved the readability of the manuscript. Tables are OK in the main text.
Answer:
Thank you very much, we have moved the images in the supplementary files section and kept the tables in the main manuscript
- Additional spell-checking and style corrections are necessary (thе problems marked in the review were in no way an exhaustive list).
Answer:
Thank you for the recommendations given. We apologize for not thoroughly checking for grammatical errors. The authors have tried to modify the additional corrections for spelling errors and grammatical errors given. The revised result is highlighted in yellow.
- Finally, I still would recommend a splitting of the text into "Bioactive compounds..." and "Antioxidant properties..." since not much was done to improve the antioxidant part in ch. 4.
Answer:
Thank you for the recommendations given to separate the Bioactive Compounds and Antioxidant Properties section. However, after discussing with all the authors, we prefer to combine the two data because these bioactive compounds emphasize that these compounds have activities, one of which is antioxidant. So, with the data on antioxidant activity, it can be confirmed that the compounds contained in the Piper species have antioxidant activity. Moreover, if these two data are separated there is no discussion that can be focused on because bioactive compounds themselves have a broad discussion as well as antioxidants. Therefore, the flow of this paper first discusses the bioactive compounds from Piper species and then more specifically examines one of the activities of these bioactive compounds in this case antioxidant activity. Hopefully our opinion can be accepted.